# Assessing the Effects and Challenges of Total Hip Arthroplasty before Pregnancy and Childbirth: A Systematic Review

**DOI:** 10.3390/jfmk9020063

**Published:** 2024-04-04

**Authors:** Athanasios Galanis, Stefania Dimopoulou, Panagiotis Karampinas, Elias Vasiliadis, Angelos Kaspiris, Evangelos Sakellariou, Christos Vlachos, Michail Vavourakis, Eftychios Papagrigorakis, Vasileios Marougklianis, Georgios Tsalimas, Dimitrios Zachariou, Christos Patilas, Iordanis Varsamos, Ioannis Kolovos, John Vlamis

**Affiliations:** 3rd Department of Orthopaedic Surgery, National & Kapodistrian University of Athens, KAT General Hospital, 14561 Athens, Greece

**Keywords:** total hip arthroplasty, hip replacement, pregnancy, pregnancy before hip arthroplasty, hip, hip resurfacing, metal on metal, childbirth, metal ions, hip arthroplasty complications

## Abstract

Total hip arthroplasty is indubitably one of the most performed operations worldwide. On the other hand, especially in the western world, the average age that women get pregnant has raised confoundedly. Consequently, a steadily increasing number of women become pregnant after they had hip arthroplasty surgery, with copious potential implications. The amount of knowledge on this particular field is considered inadequate in the existing literature. This paper aims to augment clinicians understanding surrounding this topic. A systematic literature review was conducted in accordance with the PRISMA guidelines. Papers from various computerized databases were scrutinized. Article selection was carried out by three authors independently employing specific pre-determined inclusion and exclusion criteria, while disagreements were elucidated with the contribution of other authors. A patently limited number of research articles were detected from our rigorous literature review, with only 12 papers meeting the inclusion criteria. The vast majority of studies were small-scale and examined confined population groups. Most studies had been performed in Finland, utilizing data from nationwide registries. Women with previous history of total hip arthroplasty feature increased rates of c-section delivery, although vaginal labor can be attempted with certain precautions. Hip implants’ survival does not appear to be affected from gestation, which is predominately well-tolerated from these women. Metal ion circulation in mothers’ blood has not been proven to trigger substantial complications concerning either mothers or offspring. It can be considered safe for women with such medical history to get pregnant; however, further multinational studies and pertinent research on this field are vital to attain more solid inferences.

## 1. Introduction

End-stage hip osteoarthritis in comparatively young patients aged under 45–53 years old ordinarily derives from an identifiable cause, in comparison with degenerative arthritis that routinely occurs later in life. These reasons can be dysplasia of the hip (DDH), childhood deformities such as Perthes disease, femoro-acetabular impingement (FAI), traumatic hip joint injury, avascular necrosis (AVN), infection and inflammatory arthritis such as rheumatoid arthritis (RA) [1,2]. Young patients require assiduous clinical evaluation and surgical treatment, as an increased risk of failure has been reported owing to these patients’ high demands and the expanded utilization period of the implants employed in total hip arthroplasty (THA) surgery contrasted to older patients. Additionally, when a deformity is present and previous surgeries have been executed, total hip replacement surgery can end up much more exacting and intricate [1,2].

Broadly speaking, two surgical techniques have been developed for the treatment of end-stage osteoarthritis. These are, first, THA, which is the conventional, safe and efficacious surgical procedure that has been employed in the vast majority of cases in recent decades, and secondly, hip resurfacing arthroplasty (HRA), which is a much more technically demanding and time-consuming surgical technique, but it features the advantage of considerably more bone preservation that can turn out to be pivotal in younger patients who are at higher risk of revision surgery [1,2]. In particular, THA has exhibited 10-year implant survival rates of more than 90–95% among middle-aged and elderly patients; however, these figures are lower among male patients younger than 55 years [3]. As a result, contemporary metal-on-metal hip resurfacing arthroplasty (MMHRA) could be regarded as a vital alternative for younger patients, taking into account its better long-term survival rates [3,4]. Until recently, MMHRA was contraindicated for women due to the requirement for smaller implants, but it has also been reported that this procedure can be beneficial for both men and women [4].

On the other hand, metal-on-metal total hip arthroplasty (MoM-THA) and MMHRA remain contentious for women with end-stage osteoarthritis also because of the potential detrimental effect of metal ion levels on the fetuses of pregnant women. Nonetheless, according to current research, it seems that the clinical importance of the existence of metal ions in umbilical cord blood is negligible, although further pertinent investigations should be carried out on a larger scale [5].

It is an indubitable fact that nowadays the number of women that have undergone total hip arthroplasty operations and have not yet gave birth or completed family planning is steeply soaring. Consequently, a new trend has emerged concerning examining the potential effects and associations of THA conducted before pregnancy and childbirth [6,7]. More specifically, it has become rapidly common for women aged over 35 years old in the Western world to defer childbearing because of the aspiration to build a felicitous professional career, obtain financial steadiness and in many cases owing to the absence of a supporting partner. When they do decide to proceed with bearing a child, it is crucial to underline that these women face a confoundedly enhanced risk of pregnancy-associated complications such as high blood pressure, gestational diabetes, placenta previa, preterm birth and copious other conditions like heart diseases and obesity [6]. Apart from that, the increased incidence of these complications, paired with pre-existing medical history such as THA surgery, might require high-risk pregnancy monitoring [5,6]. At this point, it is vital to underline that designing for the whole population is an essential requirement when manufacturing any product in general. This postulate also applies to hip implant manufacturers, as they should try to consider pregnancy-related needs when designing their products [8].

Perusing the existing literature, only a narrow number of studies exists inspecting the potential pregnancy complications that might accrue after THA or HRA, which could be attributed chiefly to the limited number of participants. Besides that, only two present-day systematic reviews on this field were found. A paper by Oliveiro et al. [7] reported that there is no THA impact on the success rates of vaginal delivery and no pregnancy effect regarding the functionality of THA implants. In the case of MoM-THA or MMHRA, diligent pregnancy planning is admonished after several months, whilst in the case of ceramic or polyethylene implants, no pregnancy-related contraindications were discerned [7]. Contrary to this study, the other recent systematic review by Samuel L.T. et al., regarding the impact of metal-on-metal hip implants on pregnancy, delineated that the occurrence of metal ions of chromium and cobalt in the umbilical cord does not appear to be the source of fetal congenital malformations; hence, no pregnancy deferral should be demanded [5]. These outcomes, although further investigations are requisite, are exceedingly important for women over 35 years old, as delaying pregnancy even for a few months might trigger an increased risk of pregnancy complications, confined fertility and augmented risk of chromosomal abnormalities and genetic syndromes [5,7]. Nevertheless, the most prevalent limitation of these reviews is the small sample size of the studies included.

Our systematic literature review aims to summarize all present-day knowledge regarding the impact of total hip arthroplasties on pregnancy outcomes, including gestation rates, hip pain, delivery mode, revision rates and potential congenital defects of newborns. It is the first review to comprise all available pertinent information and to utilize broader inclusion criteria, while also involving the obstetrician’s point of view in the commentary.

## 2. Materials and Methods

An extensive and rigorous literature review was executed, scrutinizing all papers that were published until November 2023 investigating the impact of total hip arthroplasty and metal-on-metal hip resurfacing arthroplasty on women of childbearing age and their potential effect on pregnancy outcomes. Also, the contingent effect of metal-on-metal protheses on fetal development through the transplacental transfer of metal ions was examined. The review was carried out in accordance with the Preferred Reporting Guidelines for Systematic Reviews and Meta-analyses (PRISMA), and therefore no ethics approval was required.

Publications from the following computerized databases were perused: MEDLINE\PubMed, Google Scholar, EMBASE, Web of Science and Scopus. The keywords employed in multiple combinations for the literature search were as follows: “metal-on-metal”, “pregnancy”, “hip”, “total hip arthroplasty”, “hip replacement”, “metal ions”, “cobalt”, “chromium”, “resurfacing”, “arthroplasty”, “malformations” and “complications”. English was activated as language filter. In addition, no limitations were applied concerning the publication date of the scientific articles. Article selection was performed independently by three authors, while disagreements were clarified with the assistance of two additional authors who made the final resolution.

Inclusion criteria were all studies and case reports investigating women of childbearing age whp had undergone total hip arthroplasty or hip resurfacing surgery and the effect of the arthroplasty on the subsequent pregnancy and fetal development. On the other hand, exclusion criteria were articles not written in English, studies on women having hip arthroplasty exclusively after pregnancies, duplicate studies, unrelated case reports, previous reviews and previous meta-analyses. Finally, reference lists from papers that met the inclusion criteria were further scrutinized to retrieve additional results (Figure 1).

## 3. Results

### 3.1. Early Case Reports

The first report identified during our literature research that explored the pregnancy outcome of a woman who had undergone total hip replacement, was a case report published in 1982 [9]. The patient was a woman suffering from juvenile rheumatoid arthritis, who had undergone bilateral total hip arthroplasty approximately one year prior to pregnancy. Throughout her gestation, she was on medication with aspirin and antibiotic therapy. During the third trimester, the patient was mobilized utilizing canes and crutches when it was requisite, while she delivered by c-section owing to a past medical history of a previous one. At the postpartum evaluation 6 weeks post-delivery, the patient was symptom-free and did not require any notable succor. The second case report germane to our study was published back in 1987, describing the case of a female patient who underwent total hip arthroplasty due to bilateral hip dislocation [10]. This was the first published paper of a patient with bilateral hip arthroplasty who delivered vaginally. Epidural anesthesia was considered not a prudent option, owing to the risk of the patient moving her legs outside of the normal range of motion, which could result in implant-loosening or dislocation. That article also highlighted the lower risk of infection compared to delivery via c-section.

### 3.2. First Research Examining THA Impact and Later Larger Scale Studies

Many years later, McDowell et al. in 2001 performed a small-scale study of five women delivering successfully after a total hip arthroplasty [11]. The objective of that study was to survey the pregnancy outcomes after hip arthroplasty as well as the functionality of the prothesis. Three out of six deliveries were executed by c-section for obstetric causes, whilst vaginal deliveries were conducted in the lithotomy position. No pregnancy complications were detected with regard to hip implants. Also, in terms of the assessment of the impact of pregnancy on the performance of the prosthesis, McDowell et al. employed the Harris hip score (which is widely considered the most ordinarily used score concerning hip capacity) in the group of women that exhibited felicitous pregnancies after hip arthroplasty and juxtaposed these results to the scores of eight non-pregnant women who had undergone total hip arthroplasty. The figures concerning the first group were 94 prior to pregnancy and 97 at the time of the most recent follow-up, while for the second group, they were 91 one or two years after the hip arthroplasty and remained unaltered at the most recent follow-up. In addition, the arc of hip motion was 217° before pregnancy and 241° at the time of the most recent follow-up in terms of the first group, whilst regarding the other group, it was calculated at 193° one to two years after hip arthroplasty and 190° at the time of the most recent follow-up. Thus, a substantial discrepancy regarding the arc of hip motion was descried among the two groups at the latest follow-up. Moreover, femoral osteolysis aspects were noted during the radiographic evaluation in three hips in both groups.

Another study by Meldrum et al. [12] utilized the Harris hip score on 107 women who were evaluated after total hip arthroplasty performed on one or two hips. The average modified score was 82.2 at the time of the follow-up, which was the same 10 years after the operation. The most engrossing aspect of the study is that 13 women involved in that study (12%) completed one or more pregnancies after the initial hip arthroplasty operation. These women featured much higher Harris hip scores compared to the others, with the figures not diverging regardless of the mode of delivery. More specifically, 13 women had 11 vaginal deliveries, 5 scheduled c-sections and 3 unarranged c-sections. The indication for four out of five planned c-sections was the previous hip surgery. Contrariwise, the indications for the remaining c-sections were privily obstetric reasons, although the history of previous hip surgery was notably taken into account. Overall, no complications in terms of hip functionality were delineated after pregnancy and childbirth. These findings correspond with the outcomes from another study by Lally et al. [13], who employed the Western Ontario and McMaster Universities Arthritis Index (WOMAC) pain and function score for the evaluation of pain and functional capacity pre- and post THA in 171 women. Out of these women, some reported pregnancy pre-THA, some others post THA, while a portion of them did not report pregnancy at all. The evaluation was performed 12–72 months after the hip surgery, with the post-THA pain scores being 85.2 ± 18.8 for women with no reported pregnancy, 84.9 ± 15.8 for women with pre-THA gestation and 91.1 ± 15.3 for women with post-THA gestation. Besides that, the post-THA functionality scores were 93.7 ± 6.4 for women with post-THA pregnancy, 91.1 ± 15.3 for women with pre-THA gestation and 87.6 ± 22.1 for nulliparous women. Finally, the mean pain scores were found to be 87.5 ± 13.5 and the mean capacity scores were 90.7 ± 14.6.

In another study, Yazici et al. [14] retrospectively evaluated the gestation outcomes of 21 women with implanted hip prostheses. They demonstrated no unpropitious impact on the implants that could be ascribed to pregnancy, while prior hip arthroplasty posed no risk on subsequent gestations. Regarding delivery mode, 6 out of 21 deliveries were carried out by c-section. In three cases, there was a conspicuous obstetric reason for the c-section, whilst regarding the other three cases, the rationale behind the c-section was obscure. Furthermore, a sizeable study of 343 women with 420 hip arthroplasties was conducted by Sierra et al. [15], investigating pregnancy and childbirth after THA. In particular, 13.7% of the total female population examined completed a successful gestation, delivering vaginally or by c-section. According to the results, vaginal labor is feasible in the vast majority of women with a history of hip arthroplasty, whilst concerning c-section rates for the first pregnancy, they were roughly 35%, which is broadly globally acceptable. The ground for the c-sections was predominantly obstetric causes and, only in a limited number of cases, the worry of the mother or the clinicians for the hip implants. This study denoted that vaginal delivery poses no risk of hip implant-associated intricacies. Apart from that, another extremely salient conclusion of this research is that THA revision risk appears not to be influenced by gestation or delivery mode, but the age of the patient undergoing THA seems to be the cardinal factor for determining the time of revision surgery. More specifically, 114 women who had not become pregnant and 24 women who became pregnant underwent THA revision surgery. The revision risk among the two groups was not correlated to gestation and childbirth, remaining changeless between women who delivered vaginally and by c-section. The risk of revision attributed to groin pain was 3.95 times higher in patients with persistent groin pain post-delivery. Nevertheless, during statistical analysis, after the needful modifications were implemented in terms of patients’ age, the significance of groin pain was nugatory [15].

Furthermore, in 2007, a study published by Stea et al. [16] inspected the outcomes of pregnancy and childbirth on women with past medical history of THA. Overall, 19 gestations transpired in 14 out of the 247 women who were originally discovered for the objective of the research. All subjects except for one delivered by c-section owing to a previous history of THA, and in one case, also due to breech fetus presentation. Only 4 out of 14 patients complained of hip-associated symptoms during gestation, while even featuring sports practice 0.5 times per day contrasted to 1.1 times per day of sports practice reported by women who did not deliver. Likewise, in this study, two felicitous gestations were recorded after revision THA surgery, which is ordinarily scarcely reported; yet, overall revision rates were analogous between women who achieved a successful pregnancy and women who did not [16].

### 3.3. Studies from Finland

Contrariwise to these findings, a nationwide register-based cohort study conducted in Finland demonstrated that in a population of 2.429 women who underwent THA, contrasted to a control group of 7.276 women, the rates of stillbirth were 2% after THA and only 0.3% prior to THA, the figures of small-for-gestational-age (SGA) neonates were 8.3% post THA and 3.3% before THA, whilst the rates of preterm births were 13.7% after THA and 7.1% prior to the operation [17]. Data in terms of pregnancy rates in this study were retrieved from 204 deliveries in women post THA and were collated to 1417 deliveries in the control group, revealing that there was an enhanced chance of elective c-section (33.8%) and a decreased chance of labor trial (66.2%). The findings concerning the control group were 8.8% and 91.2%, respectively. The numbers regarding vaginal deliveries for the patient group were for spontaneous vaginal delivery 68.9%, for assisted vaginal delivery 2.2% and for emergency c-section 28.9%, while in terms of the control group, the figures were 82.1%, 6.5% and 11.6%, correspondingly [17].

In addition, in a research paper scrutinizing the data that were acquired from the same Finnish nationwide registries, it was reported that THA revision rates on women after delivery were 51 revisions out 133 THA performed (in 111 women) and 645 revisions out of 2.343 THA (in 1878 women) executed in the control group [18]. More particularly, 59% of the revisions in the delivery group were ascribed to aseptic implant-loosening, contrasted to 49% of the revisions in the control group that were also carried out for the same cause. Also, out of the 170 deliveries that took place after THA, 75 were vaginal and 95 were c-sections. It is essential that revision indications did not vary among the vaginal delivery and c-section groups. Furthermore, survival rates at the 6-year follow-up were 91% in the delivery group and 88% in the control group, whilst at 13 years, they were 50% and 61%, respectively [18].

Concerning birth rates of women with a past medical history of THA, an extensive Finnish population-based study was also performed on 3.434 men and 2.429 women, comparing them to other 10.299 men and 7.276 women as a control group [19]. All data were obtained from the Finnish Arthroplasty Register. The birth rate after THA was approximately 20–60% lower in the male and female patient groups than in the reference individuals. According to the findings of the survey, birth rate was predominately determined by the number of children who were born before THA surgery, the age of the subjects who were included in the scrutiny and the relationship status. A previous medical history of diabetes mellitus or RA was also taken into account. Overall, birth rates were lower in women with a history of THA than in women of the control group. More specifically, the birth rate of women aged between 15 and 19 years old (per 10,000 person-years) was 216 for women with a THA history and 570 for women without THA performed. Besides that, the birth rate for women aged between 20 and 34 years old was 165 for those with a THA history and 400 for those without. Also, in terms of women between 35 and 39 years old, the birth rate was found to be 25 for women with a THA history and 63 for those without, while regarding women aged between 40 and 45 years old, the birth rate was 2.4 for those with executed THA and 5.4 for those without [19]. Furthermore, research on valuable data acquired from five national health registers of the Finnish population, including the Finnish Arthroplasty Register, also revealed that among 1713 women with a preceding THA history, 1274 gestations and 187 induced abortions (IAs) occurred [20]. In particular, 17.9% of IAs transpired post THA, as opposed to 14.1% before THA. The corresponding result regarding the control group was 13.9%. Finally, IA rates post THA owing to maternal health issues were 14.7%, contrasted to 2.7% in the control group [20].

### 3.4. Studies concerning Metal-on-Metal Hip Implants and Potential Effects on Newborns

Regarding research investigating the potential harmful effect of metal-on-metal THA implants on pregnancy and childbirth, a confoundedly small number of papers is available in the existing literature (Table 1). More specifically, in 2007, a controlled study by Ziaee et al. [21] examined the gestations of women with history of metal-on-metal hip implants. Blood samples from women who had undergone metal-on-metal hip resurfacing surgery and from the umbilical cord were obtained at the time of delivery, before intravenous fluid infusion and/or blood transfusion [21]. Analogous samples were acquired from women without metal implants. The primary aim of the study was to define the levels of chromium and cobalt, distinguishing potential congenital abnormalities and any correlations between them. In accordance with the findings, chromium and cobalt metal ions were discerned in maternal and umbilical cord samples in both groups of women. Higher cobalt and chromium levels were detected in the maternal blood of the study group compared to the corresponding levels of the control group. Higher cobalt levels were found in the cord blood sample of the study group than those of the control group, whereas chromium levels in the cord sample of the study group featured no statistically significant discrepancy when contrasted to the control group [21]. These outcomes are in accordance with the results from a case series report published in 2012 by de Souza et al. [22], which inquired into the cobalt and chromium levels of three women who had undergone metal-on-metal hip resurfacing surgery, confirming that the transplacental passage of metal ions was present. In particular, maternal cobalt levels were higher compared to chromium levels, while gestation prompted alterations in pelvic structure that could induce the release of increased levels of metal ions. No teratogenic effects on the newborns were observed; yet, no concrete data existed in terms of the children’s later development [22]. Finally, a study by Kuitunen et al. [23], utilizing data from the Finish nationwide registry based on 2429 women with THA history compared to 7276 women as a reference group, denoted that in the THA group, three (5.9%) out of the total number of IAs were executed owing to suspected fetal defects, while four (1.9%) stillbirths and eight (3.8%) newborns with one or more major abnormalities were noted. Regarding the control group, 13 (5.5%) out of the total number of IAs were performed due to suspected fetal defects, whilst eight (0.6%) stillbirths and 47 (3.3%) newborns with one or more major defects were recorded. No noteworthy discrepancies were descried in terms of the rates of defects between women who had undergone metal-on-metal THA and those with other types of hip prostheses [23].

## 4. Discussion

For a thorough investigation of the potential impact and challenges that THA exerts on pregnancy and childbirth and with the objective of obtaining the most solid inferences possible, it is vital to respond to some fundamental queries. These issues are the following: Is there any difference on pregnancy rates between women with and without hip implants? Are there higher hip pain levels during pregnancy in women with hip implants? Do hip implants affect the mode of delivery? Does pregnancy affect revision rates in women with hip implants? Are there any congenital defects in the newborns of women with previous THA surgery? The following subsections aspire to provide answers to these basic questions.

### 4.1. Previous Total Hip Arthroplasty or Not Having any THA and Pregnancy Rates

Consistently with the papers that are included in our study, a comparatively lower rate of gestations has been noticed in women that have previously undergone total hip replacement operations. The research performed by Artama et al. [19] demonstrated that in a population that involved both men and women with a past medical history of total hip arthroplasty surgery, there existed a much lower probability of having children after surgery. More specifically, the likelihood of not having a child after THA was found to be lower for men compared to women, while it was even higher if the person had live-born children before the operation [19]. A possible explanation for these outcomes is that people at a younger age, who are generally more likely to decide to start a family, commonly suffer from a systematic illness (such as RA) that leads to THA surgery, affecting their quality of life in general as well as their sexual well-being.

Additionally, older studies, such as Meldrum’s et al. research [12], denoted a gestation rate of 12% for the women of the study group. The women of that study who proceeded with at least one successful pregnancy were younger and did not exhibit any patent differences concerning follow-up. One patient resolved not to go on for a second pregnancy owing to her fear of vitiating the hip implants. Likewise, the pregnancy rates of the study conducted by Stea et al. [16] were among the lowest that we detected in our research (10%); however, that study was carried out in a different population which featured low fertility rates in total, and thus the even lower pregnancy rates in women with congenital hip disorders were immensely anticipated. In a prospective study by Yoon et al. [24], which examined alumina-on-alumina THAs executed in patients younger than 30 years with an enormous follow-up period, 11 out of 29 female patients in the study population carried on with at least one felicitous gestation. In our opinion, the increased pregnancy rates that were reported in this study could be attributed to the fact that the study population were patients younger than 30 years with no previous live-born children, and therefore they had considerably higher chances of deciding to start a family. Also, Sierra et al. [15] concluded that 17% of patients managed to proceed with at least one pregnancy, with the mean time from THA to the first pregnancy being 3.8 years, whilst Yazicy et al. [14] claimed that 20% of the patients included in their study exhibited successful pregnancies. Taking into consideration these findings, the gestation rates in women with a previous history of hip replacement surgery are considered low, with these figures appearing consistent in the studies that we scrutinized during a period that covers roughly the past 20 years. Recently developed surgical procedures and techniques, as well as more contemporary types of implants, do not seem to impact these outcomes [25]. In general, younger patients and patients who do not have previous live-born children are considered more likely to attempt pregnancy; yet, the overall lower quality of life and systemic disease that presumably induced the hip surgery at a young age could be blamed as the principal origin of low birth rates in this particular population.

### 4.2. Hip Pain Levels of Women with THA History during Gestation

Hip pain was unquestionably the most examined factor among the scientific articles that we perused. Meldrum et al. [12] were the first to focus their research on the women’s functional status and pregnancy-associated complications of THAs by employing the modified Harris hip score for the evaluation of the level of hip pain at the time of follow-ups. According to that study, women who became pregnant featured pronouncedly higher scores in comparison with women who did not get pregnant, while also being more functional and less contingent on pain-relief treatment. Besides that, they were reported not to complain of any pain or discomfort during gestation that could be related to the hip surgery. This could be justified by the fact that women fewer less clinical symptoms and a better prognosis decide to get pregnant and start a family more straightforwardly. Findings from the study by Lally et al. [13], which measured WOMAC scores, suggested that women exhibited similar postoperative pain regardless of the time of gestation. Contrariwise, results from the study by Sierra et al. [15], which involved a large population of women with a previous history of THA undergoing pregnancy, indicated that 60% of the studied population manifested pregnancy-related hip pain. Also, it was pointed out that persistent pain even after delivery was a material contributing factor regarding the risk of early revision and the loosening of the THA’s acetabular component [15,26]. On the other hand, the findings that were published by Stea et al. [16] connoted that 20% of the examined population reported moderate hip pain and discomfort, a figure that was not substantially dissimilar from those of women without hip implants. Based on our review and taking into account the heterogeneity of the outcomes of some papers, gestation does not appear to markedly alter hip pain levels in women with a previous history of THA; however, groin pain during pregnancy and after delivery could pose an increased risk of early THA revision. Additionally, considering the fact that medical conditions that could lead to the necessity for hip arthroplasty at an early age are ordinarily severe and occasionally systematic diseases, women who will make the decision to get pregnant are usually healthier, fitter and report fewer clinical symptoms when contrasted to the rest of the population who has undergone THA at a relatively young age.

### 4.3. Hip Implants and Mode of Delivery

Earlier studies denoted that the rates of c-sections were notably higher than the rest of the population, as anticipated for women with previous hip surgery and medical conditions. Medrum’s study [12] indicated a rate of 42% of c-sections compared to 22% which was the national average. It has to be emphasized that the vast majority of the planned c-sections were scheduled due to the hip condition. Furthermore, the study by Yazici et al. [14] recorded 29% of c-sections at the studied population, while Sierra’s study [15] detected a slightly higher rate of 36% of c-sections, which were carried out partially owing to the previous hip arthroplasty and partly due to copious obstetric reasons. Contrary to these outcomes, the research by Stea et al. [16] demonstrated a 93% c-section rate, which is regarded as an exceedingly high rate. This finding can be explained by the fact that the study was performed in Italy, which is characterized by a vastly different population group featuring a very high national rate of c-sections at 36.9%, with many obstetricians and women adhering to a low threshold to opt for delivery by c-section. Also, another rationale behind these numbers could be the high prevalence of patients with developmental hip dysplasia leading to a reduced pelvis size.

Moreover, according to the outcomes from the study by Kuitunen et al. [17], the higher rate of elective c-sections in the patient group can be interpreted by the unavoidable sentiment of anxiety and fear concerning the functionality of the implants that many female patients unwittingly experience. It needs to be stressed that these women customarily suffer from diseases like RA, congenital hip dysplasia or hip osteonecrosis. These patients should be reassured that they can proceed with vaginal delivery without expecting complications, if hip flexion is maintained below 90° [23]. Furthermore, the lower probability of labor trial success can be ascribed to the fact that many women had already arranged an elective c-section and trial was undertaken due to early labor onset.

All in all, hip implants do seem to exert influence on the mode of delivery since most c-sections are scheduled because of the previous history of hip arthroplasty. Trial of labor is obtainable and vaginal delivery is feasible if hip flexion is maintained below 90%.

### 4.4. Revision Rates in Women with Previous Hip Arthroplasty and Subsequent Pregnancy

The first large series of women who had undergone THA and then completed gestation was carried out by Sierra et al. [15], reporting that no significant discrepancies were discerned concerning revision rates between women with a previous THA who delivered vaginally or by c-section and women who had no such history. The pivotal factors that determined the survival rates of the implants were the age of the hip arthroplasty operation and the existence of symptoms such as groin pain, which was evaluated by calculating the Harris hip score. According to a research paper by Khan et al. [26], groin pain can be regarded as a potent indicator of isolated acetabular component loosening, as 89% of 57 patients with femoral component loosening and all patients with loose acetabular component in the population that was examined complained of thigh pain. In addition, the study by Medrum et al. [12] underlined the momentousness of investigating the impact of gestation on the longevity of the hip implants, firstly because these prostheses are implanted for a prolonged period of time, and secondly because pregnant patients are often young, featuring high demands and being potentially affected by serious underlining medical conditions that induced the requirement for hip arthroplasty at an early age [27]. The population examined in that research was small; however, no higher incidence of repeat surgeries or revisions was distinguished in pregnant patients compared to controls [12]. Likewise, the study conducted by Stea et al. [16] inferred that there was no considerable difference regarding revision rates between the mother and the non-mother groups. More specifically, in terms of the non-mother group, the mean survival time of the implants was 5.5 years and failure transpired preponderantly owing to cup-loosening. Concerning the mother group, only one subject required revision surgery after gestation due to ceramic liner chipping. Nonetheless, it was not explicit whether this incident occurred or aggravated due to childbirth. Complementary to these findings, the 10-year follow-up study by Yoon et al. [24] on patients with alumina-on-alumina THA, comprising 62 patients, 11 of whom manifested successful pregnancies, revealed that no revision surgery was required because of pain post delivery. Yazici et al. [14] did not report any alterations in terms of the hip implants’ survival rates in their studied population either.

Furthermore, the findings from the more recent register-based research by Kuitunen et al. [18] indicated that delivery does not appear to impact the survival rates of the hip implants in women after primary THA. In accordance with the outcomes from Sierra’s et al. paper [15], age is the only paramount element that can alter the survival of the hip implants. Additionally, a single-center Swedish study conducted by Mohaddes et al. [28] proposed that the survival of hip implants in patients aged less than 30 years old is lower compared to older patients. Likewise, other studies that examined the survival rates of hip implants in young patients demonstrated contiguous survival rates in female patients who did not deliver [29,30]. Perusing the aforementioned results, vaginal labor or c-section does not seem to affect hip implants’ survival rates; hence, women should not avoid gestation due to that reason. On the other hand, since it has been reported that the amount of daily-life activities can affect THA’s wear in the general population [31], this specific fact should also be considered. Further research is required in this field, as this aspect was not adequately examined in the studies analyzed.

### 4.5. Newborns of Women with Previous Total Hip Arthroplasty and Potential Congenital Defects and Complications

The study carried out by Kuitunen et al. [23] denoted that pregnant women with a past medical history of THA with non-MoM implants did not feature higher rates of delivering offspring with congenital anomalies in comparison with the control group. Moreover, women with MoM hip implants demonstrated a comparable risk of having offspring with congenital defects when contrasted to the control group, as the discrepancies were not statistically significant. Nevertheless, that paper highlighted the requirement for conducting further apposite research studies in order to achieve more reliable outcomes. On the other hand, there exists research suggesting that increased levels of chromium and cobalt can induce chromosomal alterations. More specifically, a study performed by Landon et al. [32] detected a six-to-seven-fold rise in chromium and cobalt levels in patients two years after MoM hip arthroplasty in comparison with metal ion levels before the hip surgery, as well as a noteworthy growth in terms of chromosomal aberrations and aneuploidy. Nonetheless, no recognized clinical consequences or long-term effects have been described. Apart from that, findings from the study by Ziaee et al. [21] corroborate that cobalt and chromium can cross the placenta in patients with and without metal implants, whilst the placenta is characterized by an adjusting impact on the metal transfer rates which are distinct for every element. It is particularly interesting that according to a prospective longitudinal study carried out by Daniel J et al. [33], examining urine and blood samples, a higher amount of circulating metal ions was noticed in patients during the first 6 months after the hip operation. Finally, it has been suggested that certain underlying conditions, such as RA, can raise the risk of preterm birth and small-for-gestational-age neonates [34], while stillbirth is also believed to be increased in patients suffering from RA [35]. These inferences were not affected by the existence or not of a previous THA. Taking into deliberation these results, we consider relatively safe to claim that metal hip implants do not have a significant impact on the fetus. Nevertheless, the necessity for continuing pertinent research in that field is unnegotiable, since outcomes are based on a small number of studies with a limited population. However, since contemporary research data suggest that the utilization of cobalt-chrome hip implants yields considerably high levels of metal ions in general, it might be advisable to suggest that females who are planning to undergo THA surgery and still have plans for childbirth avoid these implants if possible [36].

## 5. Conclusions

Following a scrupulous review of the available literature, gestation rates in women with a past medical history of total hip arthroplasty are indubitably firmly rising. Knowledge acquired from the existing studies propounds that these women feature higher rates of c-section delivery, although vaginal labor can be attempted with the assistance of an adept obstetrician who should pay particular attention to the extent of hip flexion throughout childbirth. The survival of the hip implant is not considered to be affected by pregnancy, while the whole gestation period is broadly well tolerated. However, hip or groin pain during pregnancy or post delivery could be a significant prognostic factor regarding determining the survival of the implants. Metal ions that circulate in the mothers’ blood have been proven capable of crossing the placenta; yet, no notable concrete risks for the fetus and offspring have been delineated. Included in this paper were studies examining specific populations, and therefore we reckon that it is vitally important for the scientific community to carry out multinational studies and research involving larger population groups in order to attain more precise outcomes. Lastly, it can be deduced that it is safe for women with a previous hip arthroplasty history to get pregnant, considering that the underlying condition that might have led to the operation at a young age is well monitored. 

## Figures and Tables

**Figure 1 jfmk-09-00063-f001:**
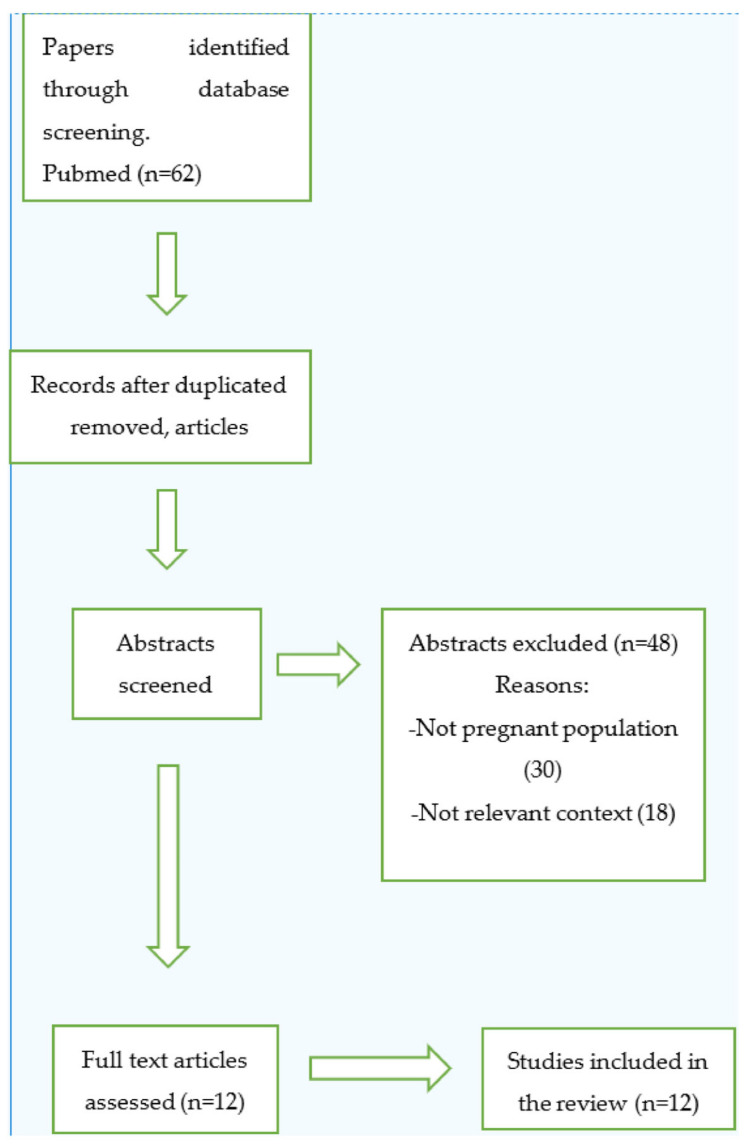
Study’s flow-chart.

**Table 1 jfmk-09-00063-t001:** Major findings of the main papers reviewed.

Reference	Title	Study Design	Results
Meldrum R et al. [12]	Clinical outcome and incidence of pregnancy after bipolar and total hip arthroplasty in young women	Retrospective study	1. Not enhanced pregnancy-associated complications and hip-related conditions. 2. Hip arthroplasty was regarded as indication for c-section. 3. Younger women with higher functional results were more likely to get pregnant.
Yazici Y et al. [14]	Pregnancy outcomes following total hip arthroplasty: a preliminary study and review of the literature	Retrospective study	1. Hip arthroplasty poses no unfavorable perils on gestation. 2. No prothesis-associated complications were detected.
McDowell CM et al. [11]	Pregnancy after total hip arthroplasty	Retrospective study	1. Felicitous gestations can transpire after THA. 2. Hip implants are not adversely impacted from pregnancy.
Sierra R J et al. [15]	Pregnancy and childbirth after total hip arthroplasty	Prospective study	1. Childbirth is not influenced by the existence of THA. 2. Gestation post THA is not correlated with a reduced survival of the implants. 3. Hip pain is ordinary during gestation in these patients. 4. Persistent groin pain after delivery customarily leads to THA revision.
Stea S et al. [16]	Safety of pregnancy and delivery after total hip arthroplasty	Retrospective study	Gestation and childbirth do not represent risk factors concerning THA survival.
Ziaee H et al. [21]	Transplacental transfer of cobalt and chromium in patients with metal-on-metal hip arthroplasty: a controlled study	Controlled study	1. Both cobalt and chromium do cross the placenta. 2. The placenta exerts a regulative impact on the extent of metal transfer.
Hyeong Jo Yoon et al. [24]	Alumina-on-alumina THA Performed in Patients Younger Than 30 Years: A 10-year Minimum Follow up Study	Prospective study	Augmented 10-year survival rate of cementless alumina-on-alumina THA in young subjects.
Lally L et al. [13]	Pregnancy Does Not Adversely Affect Postoperative Pain and Function in Women with Total Hip Arthroplasty	Cohort study	1. Gestation post THA was not related to worse postoperative pain or functional capacity. 2. No discrepancies in pregnancy outcomes or complications among women with post-THA gestation and those with pregnancy pre-THA.
Miia Artama et al. [19]	Lower birth rate in patients with total hip replacement	Cohort study	THA could induce lower birth rates in both men and women.
Ilari Kuitunen et al. [17]	Pregnancy outcome in women after total hip replacement: A population-based study	Cohort study	1. Neonates born after maternal THA feature an increased peril of preterm birth, stillbirth, small size for gestational age and low birthweight. 2. Trial of labor is more likely to result in emergency cesarean section.
Ilari Kuitunen et al. [18]	No effect of delivery on total hip replacement survival: a nationwide register study in Finland	Cohort study	Delivery does not appear to abate THA implant-survival rates.
Ilari Kuitunen et al. [23]	Congenital anomalies in the offspring of women with total hip replacement: a nationwide register study in Finland	Cohort study	THA does not seem to affect the risk of major congenital anomalies or gestations ending owing to suspected fetal abnormalities.

## Data Availability

All raw data are available to access should they be requested.

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
