# Peer review of "Assessing the Effects and Challenges of Total Hip Arthroplasty before Pregnancy and Childbirth: A Systematic Review"

_jfmk, 2024, doi:10.3390/jfmk9020063_

Round 1

Reviewer 1 Report

Comments and Suggestions for Authors

I would like to thank to authors for writing a review paper on the topic of THA prior to pregnancy and childbirth.

Designing for the whole population is a must while designing products as mentioned in this paper ‘B. S. Acar, S. Mihcin, International Journal of Human Factors Modelling and Simulation, 2010, 1, 380-389, doi: 10.1504/IJHFMS.2010.040272., You might consider suggesting that implant manufacturers should try to consider the pregnancy related need and consider this while designing the products incase they might have different needs.

Throughout the paper, there are many abbreviations which have not been given in the long form first then the name in parentheses such as WOMAC pain score , SGA etc .Please write the long form first then use it as an abbreviation. Or at least give some previous info about  this scoring so that the readers might understand better.

Related to MoM part, you might refer to this paper Dynamic computational wear model of PEEK-on-XLPE bearing couple in total hip replacements. Med Eng Phys 2023;117:104006. https://doi.org/10.1016/j.medengphy.2023.104006.>  which gives an advise on use of CoCr producing already high levels of metal ions in general and not being advised to use, so that with your results you might consider advising on people who are planning to have THA and still have plans of child birth to consider not to use CoCr type of implants if they have a choice to be on the safe side.

The discussion part starts with some sub sections. It might be good to have an initial paragraph to explain how you planned the section related to how you decided on the headings.

Also one of the most important factors affecting the wear in THA is amount of daily life activities. Depending on the activity level different amounts of wear could be detected. As mention in this paper. "Database covering the prayer movements which were not available previously." Scientific data 10.1 (2023): 276. You might have a subsection related to the activity levels, and mention about what sort of daily life activities these pregnant women were performing and if there was any difference among those who were sedentary and who were active, doing sports or walking only etc.

Please change this title in the discussion section from ‘Previous Total Hip Arthroplasty or Not and Pregnancy Rates’ to

‘ Previous Total Hip Arthroplasty or Not Having any THA and Pregnancy Rates’

In the list of the abbreviations, there are many missing abbreviations, please scan through the text and include them all .

Comments on the Quality of English Language

minor editing required and also issues with abbrevations should be resolved.

Author Response

Dear reviewer 1,

We sincerely appreciate your comments and expertise. Thank you very much for the time you spent in evaluating our paper and for your valuable feedback, aiding us in improving the quality of our article. We revised our paper according to your comments.

To begin with, we added your comment about implant-designing, suggesting that hip implants makers should try to consider pregnancy-related needs and also we cited the article you indicated. Moreover, in terms of abbreviations, we wrote the long form first and then use it as an abbreviation in many sports throughout the paper as you suggested. Furthermore, we referred to the paper you indicated regarding mom implants, while also suggesting that women planning to have a child might consider not using CoCr implants as you suggested. Also, in the discussion part, we added an itinial explanatory paragraph as you instructed. Additionally, we added the valuable information about activity levels and tha wear by also citing the article you indicated. Also, we accentuated the need for further pertinent research in this field as the papers involved in the review provided little information about activity levels of women after tha and pregnancy. Apart from all these, we also changed the first subtitle in the discussion part as you instructed us. Finally, we scanned through the text and included all abbreviations in the list before the references as you told us.

We really hope that the modifications we made will result in the acceptance of our paper for publication. Thank you very much again for all your precious assistance.

Yous sincerely,

The authors

Reviewer 2 Report

Comments and Suggestions for Authors

The authors have prepared a review article entitled “Total Hip Arthroplasty Prior to Pregnancy and Childbirth: Potential Impact and Challenges. A Systematic Review.” They reported as follows:

Total hip arthroplasty is a commonly performed procedure worldwide, while the average age of pregnancy in the Western world has increased significantly. This has led to a growing number of women becoming pregnant after undergoing hip arthroplasty, with potential implications. However, the existing literature on this topic is considered inadequate. This paper aims to enhance clinicians' understanding through a systematic literature review following PRISMA guidelines. Despite a rigorous search, only a limited number of research articles were found, with 12 meeting the inclusion criteria. Most studies were small-scale and focused on specific populations, primarily conducted in Finland using national registries. Women with prior hip arthroplasty may have increased rates of cesarean delivery, but vaginal labor can be attempted with precautions. Pregnancy does not appear to affect hip implant survival, and metal ion circulation in mothers' blood does not seem to pose significant risks to either mothers or offspring. While it is generally safe for women with this medical history to become pregnant, further multinational studies and research are crucial for more conclusive findings.

Considerable attention was paid to preparing the review, which is interesting. This review would show a significant impact on Total hip arthroplasty and the orthopedic community. All the sections of the manuscript were explored systematically. I would recommend the acceptance of the manuscript in its present form.

The reviewer recommends this work be considered after a major review.

1.      The title may change to “Assessing the Effects and Challenges of Total Hip Arthroplasty Before Pregnancy and Childbirth: A mini-review."

2.      The introduction section appears to be quite brief and would benefit from a comprehensive revision to help readers clearly understand the scientific problems addressed by this research. To offer a comprehensive exposition of orthopedic implants and applications, it is prudent for the authors to reference recent scholarly works authored by José M Diabb Zavala, Alex, EZ.; Mamidi, N.; Barrera, and Fatemeh Ijadi. These scholarly citations provide an invaluable foundation for probing the distinctive characteristics and merits of orthopedic implants.

Author Response

Dear reviewer 2,

We sincerely appreciate your comments and expertise. Thank you very much for the time you spent in evaluating our paper and for your valuable feedback, aiding us in improving the quality of our article.

As instructed, we changed the title of the paper. Regarding your very interesting and thought-provoking comments about the introduction, We did not added further information as the length of the article was further increased because of the comments of the other reviewers.

We really hope that the modifications we made will result in the acceptance of our paper for publication. Thank you very much again for all your precious assistance.

Yours sincerely,

The authors

Reviewer 3 Report

Comments and Suggestions for Authors

Title: Total Hip Arthroplasty Prior to Pregnancy and Childbirth: 2 Potential Impact and Challenges. A Systematic Review.

Thank you for the opportunity to review this manuscript. In this study, the authors assessed the outcomes of pregnant female patients with previous history of THA. The authors report that previous history of THA resulted in increased rates of c-section delivery. As anticipated, THA implant survival is not affected by gestation. The manuscript is well-written and adequately structured. However, there are a few remaining concerns:

-        Introduction: in the last paragraph, you should specify what exact outcomes you assessed. You can use the already established divisions in the Discussion as the axis of the study (pregnancy rates, hip pain, delivery mode, revision rate).

-        M&M: I would recommend that the authors specify the exact search query used. If the query is complex / long, it should be provided in a supplementary file. In addition, when was the search performed?

-        Results: Results are hard to follow due to the abundance of data. I recommend the authors use heading the same way they did on the discussion section.

-        Discussion and conclusions: Adequate

Overall, I liked the structure of this manuscript. My personal recommendation is to suggest publication after these issues are addressed.

Comments on the Quality of English Language

Although the grammar is correct in most cases, the English sounds funny. The authors use rather complicated sentence structures as well as abundantly use adjectives. I'd recommend having it read by a native speaker to simplify the language.

Author Response

Dear reviewer 3,

We sincerely appreciate your comments and expertise. Thank you very much for the time you spent in evaluating our paper and for your valuable feedback, aiding us in improving the quality of our article. We revised our paper according to your comments.

In terms of the introduction, in the last paragraph we specified the exact outcomes that we assessed as you instructed. Also, in the materials and methods section we added information about the timing of the literature search as you suggested. Regarding the results section, we used heading the same say as we did in the discussion section as you indicated. finally, we had our article checked by a native English speaker as you suggested.

We really hope that the modifications we made will result in the acceptance of our paper for publication. Thank you very much again for all your precious assistance.

Yours sincerely,

The authors

Reviewer 4 Report

Comments and Suggestions for Authors

This manuscript aims to summarize all present-day knowledge regarding the impact of total hip arthroplasties on pregnancy outcomes. However, there were some confusing contents in this manuscript:

 1.      Concerning pregnancy rates of females with a past medical history of THA, a Finnish population-based study was also performed on 3,434 men and 2,429 women, comparing them to another 10,299 males and 7,276 females as a control group [19]. Please change your number format. Also, note that "pregnancy rates" and "birth rates" are different concepts; you should not mistake "birth rate" for "pregnancy rate." As of the current time, there is no "pregnancy rate" for men.

 2.      In the discussion, it was stated, "More specifically, the risk was found to be lower for males compared to females, while it was even higher if the person had live-born children before the operation." Please clarify which risk was found to be lower for males compared to females.

Author Response

Dear reviewer 4,

We sincerely appreciate your comments and expertise. Thank you very much for the time you spent in evaluating our paper and for your valuable feedback, aiding us in improving the quality of our article. We revised our paper according to your comments.

In the results section, we corrected the wrong term “pregnancy rates” and used the correct term “birth rates” as you indicated. Also, in terms of the discussion section, the confusing word “risk” was rephrased appropriately as you instructed us and the meaning of the sentence was clarified.

We really hope that the modifications we made will result in the acceptance of our paper for publication. Thank you very much again for all your precious assistance.

Yours sincerely,

The authors

Round 2

Reviewer 4 Report

Comments and Suggestions for Authors

No more comments